# Coupled-Mode Parabolic Equations for the Modeling of Sound Propagation in a Shallow-Water Waveguide with Weak Elastic Bottom

Sergey Kozitskiy 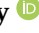

Ocean and Atmosphere Physics Division, V.I.Il'ichev Pacific Oceanological Institute, Far-East Branch of Russian Academy of Sciences, 690041 Vladivostok, Russia; skozi@poi.dvo.ru

**Abstract:** In this work, a mode parabolic equation method with interacting modes accounting for the weak elasticity at the bottom is developed. An important feature of the proposed method is that computations of elastic modes are avoided and that the solution is obtained in the form of expansion over acoustic modes. A numerical technique for solving resulting mode parabolic equations is developed, and the accuracy and efficiency of the resulting solution is validated by a direct comparison against source image solutions in the 3D wedge benchmark problem. Satisfactory agreement of the two solutions is achieved for sufficiently small values of shear wave speed that are typical for soft sediments of the sea bottom. The developed approach may be used for solving 3D problems of sound propagation with the elastic properties of bottom sediments taken into account.

**Keywords:** underwater acoustics; normal modes; multiple-scale method; elastic bottom; mode parabolic equations

## 1. Introduction

The modeling of sound propagation in three-dimensional (3D) shallow water waveguides is one of the most important areas of research in underwater acoustics [1]. Despite the impressive achievements of the past two decades, the simulation of acoustic fields in large domains with 3D inhomogeneities of various kinds remains a significant challenge, especially when broadband signals are considered. Such simulations are necessary, e.g., for the estimation of anthropogenic acoustic noise levels, which is important for the protection of marine fauna [2]. It is also important that the modeling is carried out in reasonable time (ideally, in real time). Many approaches to the simulation of 3D acoustic fields are being actively developed, including 3D parabolic equations, Gaussian beams [3], rays-theoretical techniques [4], as well as the methods based on the finite-difference or finite-element discretizations [5] of wave equations (see the nice collections of papers in recent special issues [6,7] dedicated to 3D propagation effects and techniques in underwater acoustics and references therein).

Mode parabolic equations (MPE) are a promising and convenient tool for solving 3D sound propagation problems of ocean acoustics. They have been used in adiabatic form, for instance, by Collins [8] and in the refined version by Trofimov [9]. Wide-angle and pseudodifferential MPE were also proposed in recent works [2,10] and used as a basis for the computational code AMPLE [11] that is used for computing noise levels due to broadband sources over large sea areas (e.g., during seismic survey [2]).

In the work in [12,13], this approach has been extended to the case of interacting modes and obtained very good agreement between numerical solutions by MPE and by the source image method in the wedge benchmark problem [14], which was achieved both for up-slope and cross-slope propagation.

The MPE approach involves two separate steps in calculating the acoustic field. First, we calculate the acoustic normal modes along the trace, and then we solve the system of

amplitude parabolic equations, in which we obtain the amplitudes of the acoustic modes. The summation of the modes with the calculated amplitudes gives us the desired field. The first step, the computation of acoustic normal modes in waveguides, is a delicate and important task. We should use numerical methods which provide sufficient accuracy along with a high speed of calculations. The number of such methods has been developed in the last few years. For instance, in the article [15] the Chebyshev–Tau spectral method was implemented to solve acoustic normal modes with a stratified ocean [15]. In [16] The Legendre collocation method based on domain decomposition is proposed to calculate normal modes. In [17], the algorithm using the Chebyshev–Tau spectral method was proposed to solve for the horizontal wavenumbers and modes of approximately range-independent segments. In [18], a discrete PE model using the Chebyshev spectral method is derived based on the wide-angle rational approximation. In [19], a multidomain Chebyshev collocation method for the accurate computation of normal modes in open oceanic and atmospheric waveguides was devised. In [20], a three-dimensional, coupled-mode two-way model using the direct-global-matrix technique as well as Fourier synthesis was presented.

For many practical reasons, it would be very important to generalize the MPE theory to the case of elastic bottom. Although, of course, one might perform it by computing elastic normal modes, there is an alternative and a more convenient way to take *weak* bottom elasticity into account in sound propagation problems of underwater acoustics [21]. The weak elasticity assumption significantly simplifies the derivation of the equations, but the equations obtained in this way can be applicable in practical problems when the shear wave velocity does not exceed the value of about 300 m/s. This may be true for liquid sediments and for sand or clay.

Various parabolic equations for elastic media have been derived in some articles (see [22,23] etc.). As a rule, they have different restrictions and an insufficient amount of test examples. Previously, we have derived an adiabatic mode parabolic equation taking into account the weak elasticity of the bottom [24]. It can be used in the seismoacoustics of liquid sediments when the shear modulus is small.

In this work, we derive a system of parabolic equations with interacting modes, taking into account the small shear modulus at the bottom and the only interface between the water column and elastic bottom. The derivation is based on the so-called generalized multiscale expansions method [25]. The obtained equations are numerically solved by the Crank–Nicolson scheme with iterations and are included into the software package developed for the modeling of acoustical fields using the MPE approach [13]. Then, we compare numerical results obtained by the derived elastic MPE with the source image solutions and validate the efficiency and accuracy of the equations.

## 2. Basic Equations and Expansions

We consider the propagation of time-harmonic acoustic waves in the three-dimensional waveguide $\Omega = \{(x,y,z)|0 \leq x \leq \infty, -\infty \leq y \leq \infty, -H \leq z \leq 0\}$ with weak elastic bottom, described by the linear elasticity equations [26]

$$-\rho\omega^2 u_k = \sum_{j=1}^{3} \frac{\partial \sigma_{kj}}{\partial x_j}, \quad k = 1, 2, 3, \tag{1}$$

where $\omega$ is the angular frequency of the waves, and the stress tensor of elasticity $\sigma_{ij}$ is (here $u_1 = u, u_2 = v, u_3 = w$)

$$\sigma_{ij} = \lambda \left( \sum_{k=1}^{3} \frac{\partial u_k}{\partial x_k} \right) \delta_{ij} + \mu \left( \frac{\partial u_i}{\partial x_j} + \frac{\partial u_j}{\partial x_i} \right).$$

Here, $\lambda$ and $\mu$ are the Lame coefficients, $\rho$ is the density of the medium, $u$, $v$, and $w$ are displacement components in the Cartesian coordinates with the $z$-axis directed upwards.

We postulate the following expansions of parameters and displacements

$$E_{\text{ef}} = \lambda + 2\mu, \quad E_{\text{ef}} = E_0(X,z) + \epsilon E_1(X,Y,z),$$

$$\mu = \epsilon \mu_1(X,Y,z), \quad \frac{1}{\rho} = \gamma_0(X,z) + \epsilon \gamma_1(X,Y,z),$$

$$
\begin{aligned}
u &= [u_0(X,Y,z) + \epsilon u_1(X,Y,z) + \ldots] e^{i\theta(X)/\epsilon} \\
&\quad + u_S(X,Y,z,\eta) e^{i\theta(X)/\epsilon}, \\
v &= \epsilon^{1/2} [v_{1/2}(X,Y,z) + \epsilon v_{3/2}(X,Y,z) + \ldots] e^{i\theta(X)/\epsilon} \\
&\quad + v_S(X,Y,z,\eta) e^{i\theta(X)/\epsilon}, \\
w &= [w_0(X,Y,z) + \epsilon w_1(X,Y,z) + \ldots] e^{i\theta(X)/\epsilon} \\
&\quad + w_S(X,Y,z,\eta) e^{i\theta(X)/\epsilon}.
\end{aligned}
\tag{2}
$$

Effective Young's modulus is $E_{\text{ef}} = \rho c^2$, where $c$ is the velocity of compressional sound waves [27]. Shear modulus is $\mu = \rho c_s^2$, where $c_s$ is the velocity of shear waves. We introduce slow variables $X = \epsilon x$ and $Y = \epsilon^{1/2} y$, where small parameter $\epsilon$ is the ratio of the wavelength to a typical size of horizontal inhomogeneities of the propagation medium. We also introduce fast variable $\eta = \xi(X,Y,z)/\sqrt{\epsilon}$ and assume that displacements in shear waves $u_S, v_S, w_S$ also depend on this variable $w_S = w_S(X,Y,z,\eta)$, etc.

The Ansatz (2) is derived from a general form of expansion of the displacements by considering expressions at different powers of $\epsilon$ in the basic equations (see Appendix A).

Using the notation introduced above, we expand the elasticity Equation (1). Then, we collect terms with the same powers of $\epsilon$. For $\epsilon^0$, we have the following equality

$$
\begin{aligned}
\omega^2 u_0 &= -\gamma_0 E_0 i\theta_X (w_{0z} + i\theta_X u_0), \\
\omega^2 w_0 &= -\gamma_0 [E_0(w_{0z} + i\theta_X u_0)]_z.
\end{aligned}
$$

Let us call $P = -E_{\text{ef}}(u_x + v_y + w_z)$ quasi-pressure. Performing a multi-scale expansion of $P$ (see [25]), we obtain the following expansion for this quantity $P = (P_0 + \epsilon P_1 + \ldots) e^{i\theta(X)/\epsilon}$, where $P_0 = -E_0(w_{0z} + i\theta_X u_0)$. Then,

$$(\gamma_0 P_{0z})_z + \left[ \frac{\omega^2}{c_0^2} - (\theta_X)^2 \right] \gamma_0 P_0 = 0, \tag{3}$$

where $c_0(X,z) = \sqrt{\gamma_0 E_0}$ is the respective zero-order approximation of the sound speed. Next, $P_0$ is sought in the form $P_0 = A(X,Y)\phi(X,z)$, and a Sturm–Liouville (SL) problem is obtained for $\phi(X,z)$ (called acoustic spectral problem) with $k^2 = (\theta_X)^2$ as a spectral parameter. The interface conditions for this SL problem are discussed in Section 4, and the pressure-release boundary conditions have the form

$$(\gamma_0 \phi_{0z})_z + \left( \frac{\omega^2}{c_0^2} - k^2 \right) \gamma_0 \phi_0 = 0, \tag{4}$$

It is known that such an SL problem has a countable set of solutions $(k_j^2, \phi_j)$ indexed by $j = 1, 2, \ldots$, where eigenfunctions $\phi_j$ are orthogonal to each other and can be normalized as

$$\langle \phi_i, \phi_j \rangle = \int_{-H}^{0} \gamma_0 \phi_i \phi_j \, dz = \delta_{ij}. \tag{5}$$

Each eigenfunction corresponds to a particular solution $P_{0j} = A_j(X,Y)\phi_j(X,z)$ of the equation for $P_0$.

Zero-order approximations for the displacements can be also expressed via $P_0$ (index $j$ is omitted):

$$u_0 = \frac{\gamma_0}{\omega^2} i k P_0 , \quad v_{1/2} = \frac{\gamma_0}{\omega^2} P_{0Y} , \quad w_0 = \frac{\gamma_0}{\omega^2} P_{0z} .$$

Consider now the equations at $\epsilon^1$. The first-order approximation for the quasi-pressure has the form

$$P_1 = -E_0(u_{0X} + v_{0Y} + w_{1z} + iku_1) - E_1(iku_0 + w_{0z}) ,$$

Additionally, the respective equations for the displacement components $u_1$ and $w_1$ write as

$$\omega^2 u_1 = ik\gamma_0 P_1 + ik\gamma_1 P_0 + \gamma_0 P_{0X}$$
$$- \frac{ik}{\omega^2} \gamma_0 [2\gamma_0 \mu_{1z} P_{0z} + (\gamma_{0z} \mu_1 P_0)_z] ,$$
$$\omega^2 w_1 = \gamma_0 P_{1z} + \gamma_1 P_{0z} - \frac{k^2}{\omega^2} \gamma_0 P_0 (2\gamma_0 \mu_{1z} + \gamma_{0z} \mu_1) .$$

Expressing all involved quantities in terms of $P_1$ and $P_0$, we obtain in the usual manner at the $O(\epsilon^1)$ the boundary value problem for $P_1$, which is not always solvable.

$$(\gamma_0 P_{1z})_z + \left( \frac{\omega^2}{c_0^2} - k^2 \right) \gamma_0 P_1 = k^2 \gamma_1 P_0 - (\gamma_1 P_{0z})_z$$
$$- 2ik\gamma_0 P_{0X} - \gamma_0 P_{0YY} - ik\gamma_{0X} P_0 - ik_X \gamma_0 P_0 \qquad (6)$$
$$+ \frac{E_1 \omega^2}{E_0 c_0^2} \gamma_0 P_0 + \frac{k^2}{\omega^2} \left[ 2(\gamma_0^2 \mu_{1z})_z + \gamma_{0z}^2 \mu_1 \right] P_0 = F .$$

## 3. WKB Solutions for Shear Waves

It is interesting to note that expressions for the displacements in the shear waves in WKB approximation are obtained automatically in the frame of the used multi-scale method. In comparison with the usual MPE, when we use parabolic scaling with the asymptotic variables $X = \epsilon x$ and $Y = \sqrt{\epsilon} y$ and phase $\zeta = \theta(X)/\epsilon$ in the case of the weak elastic media, we should introduce one more phase variable.

Consider the wave equation for the vertically propagating shear wave in the media with constant $c_s$: $u_{zz} + (\omega^2/c_s^2)u = 0$. However, $c_s^2 = \mu/\rho$ and $\mu = \epsilon\mu_1$. So, we obtain equation $\epsilon u_{zz} + (\omega^2 \rho/\mu_1^2)u = 0$. For this equation, we can obtain the approximate solution by the classical WKB method even without the assumption of constant $c_s$. In the multi-scale approach, we can remove $\epsilon$ from this equation by introducing the new variable $\eta = z/\sqrt{\epsilon}$.

In our more general case, when $c_s$ and other parameters are not constants, we should introduce the phase variable $\eta = \xi(X, Y, z)/\sqrt{\epsilon}$. Thus, in our considerations, only the powers of $\sqrt{\epsilon}$ take parts in the asymptotic variables. Then, it is reasonable to seek solutions for the displacements $u$, $v$, and $w$ as the asymptotic series in the power of $\sqrt{\epsilon}$. Additionally, dependent variables should depend on $X$, $Y$, $z$, $\zeta$, and $\eta$. Successive consideration of the terms of these series is performed in the Appendix A.

For the shear waves displacements, we use Expansion (A3) obtained as a result of the considerations in the Appendix A.

$$u_S = \epsilon^{1/2} u_{S1/2}(X, Y, z, \eta) + \epsilon u_{S1}(X, Y, z, \eta) + \dots ,$$
$$v_S = \epsilon v_{S1}(X, Y, z, \eta) + \epsilon^{3/2} v_{S3/2}(X, Y, z, \eta) + \dots ,$$
$$w_S = \epsilon w_{S1}(X, Y, z, \eta) + \epsilon^{3/2} w_{S3/2}(X, Y, z, \eta) + \dots .$$

Let us substitute them into the elasticity Equation (1) and collect terms at the same powers of $\epsilon$. Note that subscript $j$ is omitted everywhere in this section.

At $\epsilon^{1/2}$ in the equation for $u_S$ and at $\epsilon^0$ in the equation for $w_S$, we obtain:

$$\gamma_0\mu_1\xi_z^2 u_{S1/2,\eta\eta} + \omega^2 u_{S1/2} = 0\,, \tag{7}$$

$$\xi_z w_{S1,\eta} + iku_{S1/2} = 0\,.$$

To solve these equations, we put $u_{S1/2} = \tilde{u}(X,Y,z)\exp(i\eta)$ and $w_{S0} = \tilde{w}(X,Y,z)\exp(i\eta)$:

$$(-\gamma_0\mu_1\xi_z^2 + \omega^2)\tilde{u} = 0\,, \quad i(\xi_z\tilde{w} + k\tilde{u}) = 0\,.$$

Thus, we obtain the expression for phase variable $\xi(X,Y,z)$:

$$\xi_z^2 = \frac{\omega^2}{\gamma_0\mu_1} \Rightarrow \xi = \pm \int_{-H}^{z} q(X,Y,s)\,ds = \pm|\xi|\,,$$

where $q = \omega/\sqrt{\gamma_0\mu_1}$. Moreover, $\tilde{w} = -k\tilde{u}/\xi_z$. From the equations at $\epsilon^1$ and $\epsilon^{1/2}$ for $u_S$ and $w_S$, we obtain:

$$(2\xi_z\tilde{u}_z + \xi_{zz}\tilde{u})\mu_1 + \xi_z\mu_{1z}\tilde{u} = 0\,. \tag{8}$$

This equation with separable variables can be easily solved

$$2\frac{\tilde{u}_z}{\tilde{u}} + \frac{\xi_{zz}}{\xi_z} + \frac{\mu_{1z}}{\mu_1} = 0 \Rightarrow \tilde{u} = \frac{C(X,Y)}{\sqrt{|\xi_z\mu_1|}}\,.$$

Finally, we have solutions for $u_{S1/2}$ and $w_{S1}$:

$$u_{S1/2} = \frac{1}{\sqrt{q\mu_1}}\left(C_1 e^{i|\xi|/\sqrt{\epsilon}} + C_2 e^{-i|\xi|/\sqrt{\epsilon}}\right)\,,$$

$$w_{S1} = -\frac{k}{q\sqrt{q\mu_1}}\left(C_1 e^{i|\xi|/\sqrt{\epsilon}} - C_2 e^{-i|\xi|/\sqrt{\epsilon}}\right)\,.$$

The expressions at $\epsilon^{3/2}$ for $u_S$ and $v_S$ give us the equation for $\tilde{v}$ in $v_{S1} = \tilde{v}(X,Y,z)\exp(i\eta)$, which is identical to Equation (8) for $\tilde{u}$, and therefore we have

$$v_{S1} = \frac{1}{\sqrt{q\mu_1}}\left(C_3 e^{i|\xi|/\sqrt{\epsilon}} + C_4 e^{-i|\xi|/\sqrt{\epsilon}}\right)\,.$$

We also obtained the equation for $\tilde{u}_1$ in $u_{S1} = \tilde{u}_1(X,Y,z)\exp(i\eta)$:

$$(2\xi_z\tilde{u}_{1,z} + \xi_{zz}\tilde{u}_1)\mu_1 + \xi_z\mu_{1z}\tilde{u}_1 =$$
$$[k^2 + \xi_Y^2 + \xi_z^2(\gamma_1/\gamma_0)]\mu_1\tilde{u} - (\mu_1\tilde{u}_z)_z\,.$$

$\tilde{w}_1$ can be found from

$$\xi_z\tilde{w}_1 = i\tilde{w}_z - k\tilde{u}_1 - \xi_X\tilde{u} - \xi_Y\tilde{v}\,,$$

which also appeared when collecting terms of the order $\epsilon^{1/2}$.

## 4. Interface Conditions

We consider the only interface between the water layer and the elastic bottom. At the surface described by equation $z = h(x,y)$, we introduce the interface conditions:

$$(\mathbf{u}_+ - \mathbf{u}_-)\cdot\mathbf{n} = 0\,, \quad \mathbf{n}\cdot(\sigma_+ - \sigma_-)\mathbf{n} = 0\,, \tag{9}$$

$$\mathbf{t_1}\cdot(\sigma_-)\mathbf{n} = 0\,, \quad \mathbf{t_2}\cdot(\sigma_-)\mathbf{n} = 0\,,$$

where $\mathbf{n}$ is a unit normal vector to the surface, and the subscript '+' denotes upper limit of the respective quantities $z = z_0$ (i.e., the limit $z \to z_0 + 0$), while the subscript '−' corresponds to the limit $z \to z_0 - 0$, and $\mathbf{u}$ is the particle velocity vector.

The normal vector **n** has coordinates

$$\mathbf{n} = (-h_x, -h_y, 1) \cdot (1 + h_x^2 + h_y^2)^{-1/2},$$

while tangent vectors $\mathbf{t_1}$ and $\mathbf{t_2}$ can be written as having the components:

$$\mathbf{t_1} = (1, 0, h_x) \cdot (1 + h_x^2)^{-1/2},$$
$$\mathbf{t_2} = (-h_x h_y, 1 + h_x^2, h_y)$$
$$\times (1 + h_x^2)^{-1/2}(1 + h_x^2 + h_y^2)^{-1/2}.$$

We further assume that surface function $h(x, y)$ can be represented in the form

$$h = h_0(X) + \epsilon h_1(X, Y),$$

and therefore $h_x = \epsilon h_{0X} + \epsilon^2 h_{1X}$ and $h_y = \epsilon^{3/2} h_{0Y}$.

The interface conditions for the stress tensor modulo terms of the order $O(\epsilon^2)$ are as follows:

$$[\sigma_{33}]_- = [\sigma_{33}]_+, \quad [\sigma_{13}]_- = 0, \quad [\sigma_{23}]_- = 0. \tag{10}$$

Using the fact that the interface perturbation $\epsilon h_1$ is small, we transfer the interface conditions to the unperturbed surface $z = h_0$, retaining only the terms up to the order $O(\epsilon)$.

Thus, we obtain the following Taylor expansion:

$$\mathbf{u}(X, Y, z, \eta)|_{z=h_0+\epsilon h_1} =$$
$$\mathbf{u}(X, Y, z, \eta)|_{z=h_0} + \epsilon h_1 \mathbf{u}(X, Y, z, \eta)_z|_{z=h_0} + \dots,$$
$$\sigma(X, Y, z, \eta)|_{z=h_0+\epsilon h_1} =$$
$$\sigma(X, Y, z, \eta)|_{z=h_0} + \epsilon h_1 \sigma(X, Y, z, \eta)_z|_{z=h_0} + \dots.$$

With all the mentioned assumptions, the interface Condition (10) at $z = h_0(X)$ can be rewritten in terms of $P_0$ and $P_1$ in the following form:

$$(P_0)_+ - (P_0)_- = 0,$$
$$(P_1)_+ - (P_1)_- = -h_1(\gamma_0 P_{0z})_+ \left( \gamma_{0+}^{-1} - \gamma_{0-}^{-1} \right) \tag{11}$$
$$- \frac{2k^2}{\omega^2}(\mu_1 \gamma_0 P_0)_-,$$

$$(\mu_1 \xi_z u_{S0\eta})_- = -\left[ \frac{ik\mu_1}{\omega^2}(2\gamma_0 P_{0z} + \gamma_{0z} P_0) \right]_-, \tag{12}$$
$$(\mu_1 \xi_z v_{S0\eta})_- = -\left[ \frac{\mu_1}{\omega^2}(2\gamma_0 P_{0z} + \gamma_{0z} P_0)_Y \right]_-.$$

The interface condition for the normal displacement component from (9) is

$$[w_0 + \epsilon(w_1 - h_{0X} u_0 + h_1 w_{0z} + w_{S0})]_- =$$
$$[w_0 + \epsilon(w_1 - h_{0X} u_0 + h_1 w_{0z})]_+.$$

Let us express this condition through $P_0$ and $P_1$:

$$(\gamma_0 P_{0z})_- = (\gamma_0 P_{0z})_+,$$
$$\left[ \gamma_0 P_{1z} + \gamma_1 P_{0z} - \frac{k^2}{\omega^2} \gamma_0 P_0(2\gamma_0 \mu_{1z} + \gamma_{0z}\mu_1) \right.$$
$$\left. -ik h_{0X} \gamma_0 P_0 + h_1(\gamma_0 P_{0z})_z + \omega^2 w_{S0} \right]_-$$
$$= [\gamma_0 P_{1z} + \gamma_1 P_{0z} - ik h_{0X}\gamma_0 P_0 + h_1(\gamma_0 P_{0z})_z]_+.$$

From (7), we can express $\omega^2 w_{S0} = ik\gamma_0\mu_1\xi_z u_{S0\eta}$ and, considering the first equation in (12), we obtain $(\omega^2 w_{S0})_- = [\gamma_0\mu_1 k^2(2\gamma_0 P_{0z} + \gamma_{0z} P_0)/\omega^2]_-$, and finally obtain:

$$
\begin{aligned}
(\gamma_0 P_{0z})_+ &- (\gamma_0 P_{0z})_- = 0, \\
(\gamma_0 P_{1z})_+ &- (\gamma_0 P_{1z})_- = ikh_{0X} P_{0+}(\gamma_{0+} - \gamma_{0-}) \\
&- h_1[\{(\gamma_0 P_{0z})_z\}_+ - \{(\gamma_0 P_{0z})_z\}_-] \\
&- (\gamma_0 P_{0z})_+\left[\left(\frac{\gamma_1}{\gamma_0}\right)_+ - \left(\frac{\gamma_1}{\gamma_0}\right)_-\right] \\
&+ \frac{2k^2}{\omega^2}[\gamma_0^2(\mu_1 P_{0z} - \mu_{1z} P_0)]_-.
\end{aligned}
\tag{13}
$$

This expression contains only $P_0$ and $P_1$ variables, so we can consider a pure acoustical case with the perturbations due to shear waves.

## 5. Boundary Conditions

At the upper boundary $z = 0$, we postulate $\sigma\mathbf{n} = 0$, which is reduced to

$$
\sigma_{33} = -P_0 e^{i\theta/\epsilon} - \epsilon P_1 e^{i\theta/\epsilon} = 0.
$$

Thus, we have Dirichlet conditions or pressure release conditions $P_0 = 0$ and $P_1 = 0$. At the lower boundary $z = -H$, we can choose $\sigma\mathbf{n} = 0$, and therefore

$$
\sigma_{33} = -P_0 e^{i\theta/\epsilon} - \epsilon[P_1 - 2(k^2/\omega^2)\mu_1\gamma_0 P_0]e^{i\theta/\epsilon} = 0.
$$

Again, we have pressure release conditions for $P_0 = 0$ and $P_1 = 0$. Moreover, $\sigma_{13} = 0$ at $z = -H$ reduces to the following equality:

$$
(\mu_1\xi_z u_{S0\eta}) = -\frac{2ik}{\omega^2}(\mu_1\gamma_0 P_{0z}).
$$

In a similar way, the condition $\sigma_{23} = 0$ at $z = -H$ reduces to:

$$
(\mu_1\xi_z v_{S0\eta}) = -\frac{2}{\omega^2}(\mu_1\gamma_0 P_{0Yz}).
$$

More useful types of boundary conditions at the lower boundary for comparison of our solutions with the solutions of ASA wedge wave propagation problems are transparent-like conditions at $z = -H$, where we require

$$
u_{S0} \sim e^{-i|\xi|/\sqrt{\epsilon}}, w_{S0} \sim e^{-i|\xi|/\sqrt{\epsilon}}, v_{S0} \sim e^{-i|\xi|/\sqrt{\epsilon}},
$$

Which can be satisfied explicitly by assuming $C_1 = 0$ and $C_3 = 0$. We also assume $P_0 = 0$ and $P_1 = 0$ and introduce an absorbing layer in order to suppress reflections of compressional waves from bottom.

## 6. Elastic Mode Parabolic Equations (EMPE)

Let $\{\theta_j | j = M, \ldots, N\}$ be a set of phases. We introduce multi-mode representation following [13]:

$$
\begin{aligned}
\mathcal{P} &= \mathcal{P}_0 + \epsilon\mathcal{P}_1 + \ldots \\
&= \sum_{j=M}^{N}\left[P_{0j}(X,Y,z) + \epsilon P_{1j}(X,Y,z) + \ldots\right]e^{i\theta_j/\epsilon}.
\end{aligned}
$$

Which gives a solution for the compressional acoustic waves. Here,

$$
P_{0j} = A_j(X,Y)\phi_j(z,X),
$$

$$P_{1j} = \sum_{l=0}^{\infty} B_{jl}(X, Y)\phi_l(z, X),$$

where $B_{jl} = \langle P_{1j}, \phi_l \rangle$ (projections obtained via inner product). Eigenfunctions $\phi_j$ with $\theta_j$ are obtained from the spectral Problem (3). Amplitudes $A_j$ can be found from the solvability condition for the boundary value Problem (6) for $P_{1j}$. For this following [13], we multiply the expression for $\mathcal{P}_1$ by $\phi_l$ and then integrate the resulting equation from $-H$ to 0 by parts twice with the use of the interface Conditions (11) and (13).

$$\sum_{j=M}^{N} e^{i\theta_j/\epsilon} \int_{-H}^{0} \left[ (\gamma_0\phi_{lz})_z + \left( \frac{\omega^2}{c_0^2} - k_j^2 \right) \gamma_0\phi_l \right] P_{1j}\, dz$$

$$= \sum_{j=M}^{N} e^{i\theta_j/\epsilon} \int_{-H}^{0} F_j\phi_l\, dz + \sum_{j=M}^{N} e^{i\theta_j/\epsilon}$$

$$\times \left\{ \phi_l \left[ (\gamma_0 P_{1jz})_+ - (\gamma_0 P_{1jz})_- \right] \right.$$

$$\left. -\gamma_0\phi_{lz}\left[ (P_{1j})_+ - (P_{1j})_- \right] + (k_j^2 - k_l^2)B_{jl} \right\}.$$

The left part of this equality turns zero due to (3), and the right part represents the desired parabolic equation. The terms $(k_j^2 - k_l^2)B_{jl}$ in these expressions can be omitted because of the resonant condition $|k_j - k_l| = O(\epsilon)$ [13].

As a result, we obtain a system of parabolic wave equations for $l = M, \ldots, N$

$$2ik_l A_{l,X} + ik_{l,X}A + A_{l,YY} + \sum_{j=M}^{N} \beta_{lj} A_j \exp(\theta_{lj}) = 0, \tag{14}$$

where $\theta_{lj} = i(\theta_l - \theta_j)/\epsilon$, and $\beta_{lj}$ is given by the formula

$$\beta_{lj} = \int_{-H}^{0} \gamma_0 \nu \phi_j \phi_l\, dz + \int_{-H}^{0} \gamma_1 \left( n_0^2 - k_j^2 \right) \phi_j \phi_l\, dz$$

$$- \int_{-H}^{0} \gamma_1 \phi_{jz}\phi_{lz}\, dz - ik_j(C_{lj} - C_{jl})$$

$$- \frac{k_j^2}{\omega^2} \int_{-H}^{0} [2(\gamma_0^2\mu_{1z})_z + \gamma_{0z}^2\mu_1]\phi_j\phi_l\, dz$$

$$+ \left\{ h_1\phi_j\phi_l \left[ k_j^2(\gamma_{0+} - \gamma_{0-}) - \left( n_0^2\gamma_0 \right)_+ + \left( n_0^2\gamma_0 \right)_- \right] \right.$$

$$- h_1\gamma_0^2\phi_{jz}\phi_{lz}\left( \gamma_{0+}^{-1} - \gamma_{0-}^{-1} \right)$$

$$\left. - \frac{2k_j^2}{\omega^2}\gamma_0^2[\mu_1(\varphi_l\varphi_j)_z - \mu_{1z}\varphi_l\varphi_j] \right\}\Bigg|_{z=h_0}.$$

Here, the values $\nu$ and $n_0$ are the same as in the study [24], coefficients $C_{lj}$ have been obtained in the work [13], and $\beta_{lj}$ differs from $\alpha_{lj}$ from the latter paper only by two terms containing $\mu_1$. Thus, in the case of low shear velocities, the problem reduces to the acoustic case with the correction in the form of these terms.

## 7. Initial-Boundary Value Problem for the System of Elastic Mode Parabolic Equations

We refer to the quantities and variables $X$, $Y$, $\nu$, $\gamma_0$ $\mu_1$, $\theta_j$, $u_{S1/2j}$, $v_{S1j}$, $w_{S1j}$, and $A_j$ as the *asymptotic* ones. Considerations of initial-boundary value problems in a partially bounded domain require the use of *physical* quantities and variables, which are $x$, $y$, $\bar{\nu} = \epsilon\nu$, $\bar{\gamma} = \gamma_0$ $\bar{\mu} = \epsilon\mu_1$, $\bar{u}_{Sj} = \epsilon^{1/2}u_{S1/2j}$, $\bar{v}_{Sj} = \epsilon v_{S1j}$, $\bar{w}_{Sj} = \epsilon w_{S1}$, $\bar{A}_j(x, y) = A_j(\epsilon x, \sqrt{\epsilon}y) = A_j(X, Y)$,

and $\bar{\theta}_j = \int^X \frac{1}{\epsilon} \theta_{j,X} \, dX = \int^x k_j \, dx$, $\bar{c}_s = \sqrt{\bar{\gamma}\bar{\mu}}$. It can be easily verified that Equation (14) in physical variables are read as

$$2ik_l \bar{A}_{l,x} + ik_{l,x} \bar{A}_j + \bar{A}_{l,yy} + \bar{\beta}_{ll} \bar{A}_l + \sum_{j=M, j\neq l}^{N} \bar{\beta}_{lj} \bar{A}_j \exp(\bar{\theta}_{lj}) = 0, \tag{15}$$

where $\bar{\beta}_{lj}$ are computed by the same formulae as $\beta_{lj}$, with $\nu$ replaced by $\bar{\nu}$, $\bar{\theta}_{lj} = (\bar{\theta}_l - \bar{\theta}_j)$.

For Equation (15), we consider the initial-boundary value problem in the domain of the form $\{(x,y) | 0 \leq x < \infty, -y_0/2 \leq y \leq y_0/2\}$, with the initial condition

$$\bar{A}_j(0,y) = g_j(y) = (g, \phi_j) = \int_{-H}^{0} \gamma_0 g(z,y) \phi_j(z) \, dz, \tag{16}$$

and the transparent boundary conditions at $y = -y_0/2$ and $y = y_0/2$ (i.e., requiring that only outgoing waves are present there). In practice, this condition is ensured by the use of perfectly matched layers.

Now, we formulate the interface conditions for the displacements in the case of the absorbing conditions at the bottom.

$$(u_{S0j})_- = \frac{A_j k_j}{\omega^3} \left[ \sqrt{\gamma_0 \mu_1} \left( 2\gamma_0 \phi_{jz} + \gamma_{0z} \phi_j \right) \right]_- \tag{17}$$

$$(v_{S0j})_- = \frac{A_j \gamma}{\omega^3} \left[ \sqrt{\gamma_0 \mu_1} \left( 2\gamma_0 \phi_{jz} + \gamma_{0z} \phi_j \right) \right]_- \tag{18}$$

$$(w_{S0j})_- = \frac{A_j k_j^2}{\omega^4} \left[ \gamma_0 \mu_1 \left( 2\gamma_0 \phi_{jz} + \gamma_{0z} \phi_j \right) \right]_- \tag{19}$$

We have expressions for the displacements for the considered case:

$$u_{S0j} = C_2 \left( \frac{\gamma_0}{\omega^2 \mu_1} \right)^{1/4} \exp(-i\eta), \quad v_{S0j} = C_4 \left( \frac{\gamma_0}{\omega^2 \mu_1} \right)^{1/4} \exp(-i\eta),$$

$$w_{S0j} = k_j C_2 \left( \frac{\gamma_0^3 \mu_1}{\omega^6} \right)^{1/4} \exp(-i\eta), \quad \eta = \frac{1}{\epsilon} \int_{-H}^{z} \frac{\omega}{\sqrt{\gamma_0 \mu_1}} \, dz.$$

Final expressions for the displacements in the shear waves in the physical variables are

$$\bar{u}_{S0j} = \frac{\bar{A}_j k_j}{\omega^2} \Phi(z,h) \left( \frac{\bar{\gamma}}{\bar{c}_s} \right)^{1/2} \left[ \sqrt{\bar{\gamma}} \bar{c}_s^{3/2} \left( 2\phi_{jz} + \frac{\bar{\gamma}_z}{\bar{\gamma}} \phi_j \right) \right]_-,$$

$$\bar{v}_{S0j} = \frac{\bar{A}_{j,y}}{\omega^2} \Phi(z,h) \left( \frac{\bar{\gamma}}{\bar{c}_s} \right)^{1/2} \left[ \sqrt{\bar{\gamma}} \bar{c}_s^{3/2} \left( 2\phi_{jz} + \frac{\bar{\gamma}_z}{\bar{\gamma}} \phi_j \right) \right]_-,$$

$$\bar{w}_{S0j} = \frac{A_j k_j^2}{\omega^3} \Phi(z,h) (\bar{\gamma} \bar{c}_s)^{1/2} \left[ \sqrt{\bar{\gamma}} \bar{c}_s^{3/2} \left( 2\phi_{jz} + \frac{\bar{\gamma}_z}{\bar{\gamma}} \phi_j \right) \right]_-,$$

$$\Phi(z,h) = \exp \left( i \int_{z}^{-h(x)} \frac{\omega}{\bar{c}_s} \, dz \right).$$

## 8. Numerical Examples

The developed model based on EMPE (15) can handle any waveguides where horizontal changes of bathymetry and hydrology are slow in comparison with the wavelength. Nevertheless, for the test calculations, it is convenient to choose a sufficiently simple waveguide with the elastic bottom for which the exact solutions, or solutions, obtained by essentially different methods are known. In our opinion, the most popular of such waveguides is the ASA (Acoustical Society of America) wedge benchmark, for which we

can obtain solutions for the acoustic fields by the source image method, even in the case of elastic bottom.

In order to illustrate the accuracy of the solution obtained using the equations derived in this study, we perform a numerical simulation of sound propagation in the standard ASA wedge benchmark problem (Figure 1) with the bottom slope angle $\approx 2.86°$ [1,14]. Moreover, in some calculations, we used a wedge with an angle $\approx 1.14°$. The sound speed in the water is 1500 m/s. The sound speed at the bottom is 1700 m/s. The bottom density is 1500 kg/m$^3$, while the water density is 1000 kg/m$^3$. We assume that the seawater is a lossless medium, while at the bottom the attenuation is 0.5 dB/$\lambda$. The point source with frequency 25 Hz is located at the depth of 100 m, and the depth of the receiver is 30 m. The water depth is 200 m at the source location. For calculation purposes, we truncate the computational domain at the depth of 1500 m below the sea surface. Shear speed at the bottom in the numerical experiments considered below is varied in the range $c_s = 0$–500 m/s. The attenuation of the shear waves at the bottom is assumed to be zero.

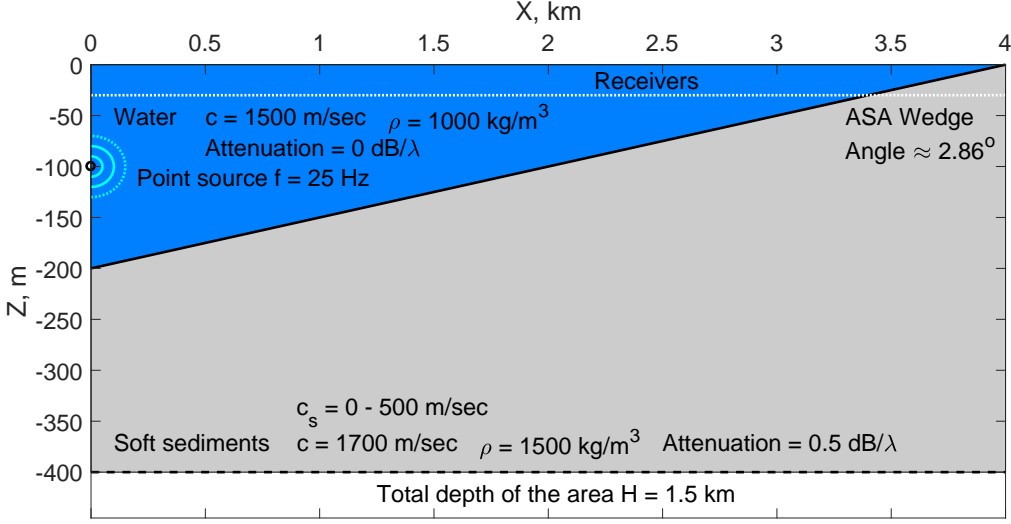

**Figure 1.** ASA wedge with the wedge angle $\approx 2.86°$.

Figure 2 shows transmission loss in the $x-z$ plane obtained by the EMPE method and the first 10 modes in this case at distance $x = 1000$ m from the source.

Figure 3 illustrates the comparison between transmission losses obtained by the EMPE method and by the source images method for $c_s = 350$ m/s (top) and $c_s = 500$ m/s (bottom). Moreover, for the case with $c_s = 350$ m/s, we additionally assumed bottom attenuation to be 0.65 dB/$\lambda$ to illustrate the effect found in computational experiments increasing the accuracy of calculations with the introduction of additional wave attenuation at the bottom. The curves for the case $c_s = 500$ m/s (bottom) illustrate how EMPE fails at large values of $c_s$. The comparison of the transmission loss obtained by EMPE with the one obtained by the source image method reveals the discrepancy of the curves at distances $x > 2$ km.

In Figure 4, we estimate the dependence of the meansquare errors of calculations from the bottom shear speed, when $c_s = 0$–500 m/s. We considered it along wedge propagation for the wedge angle $\approx 2.86°$ (left) and for the wedge angle $\approx 1.14°$ (right). One can see that in both cases of EMPE, taking into account the weak elasticity of the bottom provides better accuracy for $c_s < 270$ m/s compared with the simple MPE without such consideration.

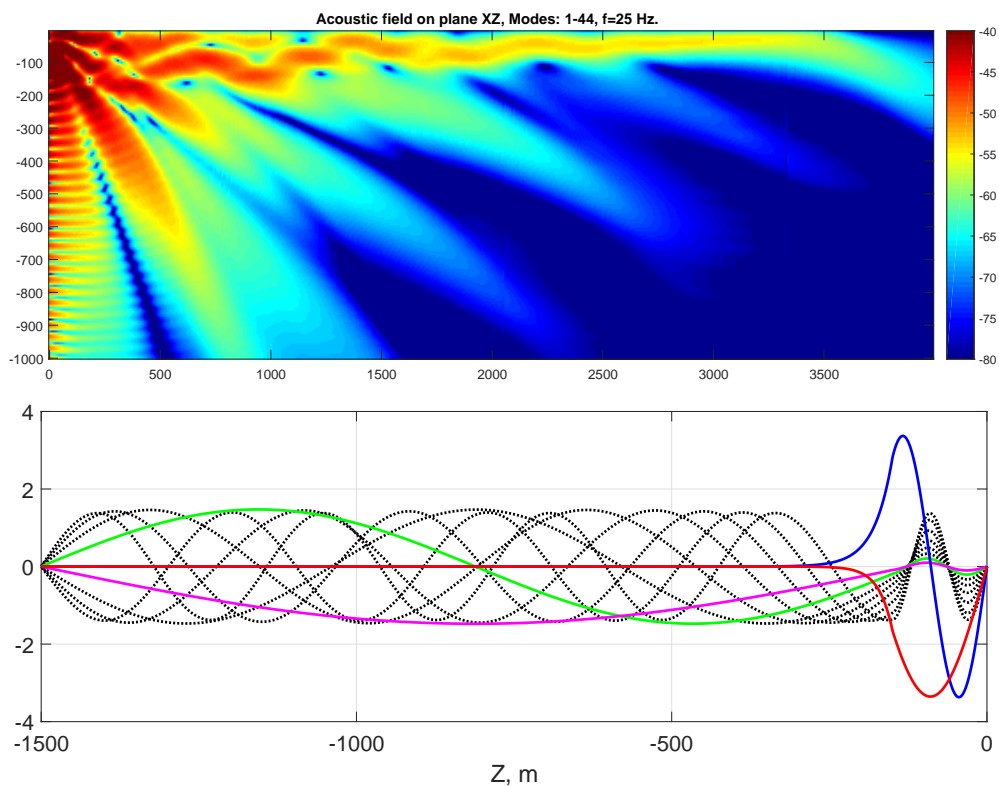

**Figure 2.** Transmission loss in the $x-z$ plane obtained by the EMPE method for $c_s = 350$ m/s and bottom attenuation 0.5 dB/$\lambda$ (**top**). The first 10 modes in this case at distance $X = 1000$ m from the source (**bottom**).

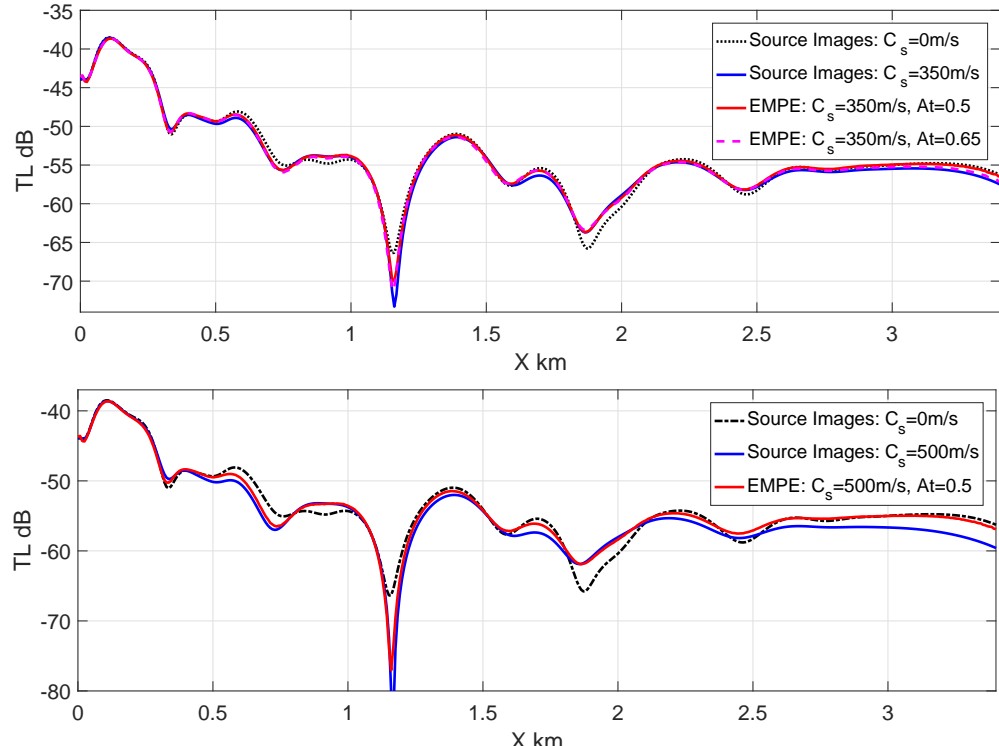

**Figure 3.** Transmission loss obtained by the elastic mode parabolic equation method compared with the source images method. Shear waves velocity $c_s = 350$ m/s (**top**) and $c_s = 500$ m/s (**bottom**), attenuation is 0.5 dB/$\lambda$.

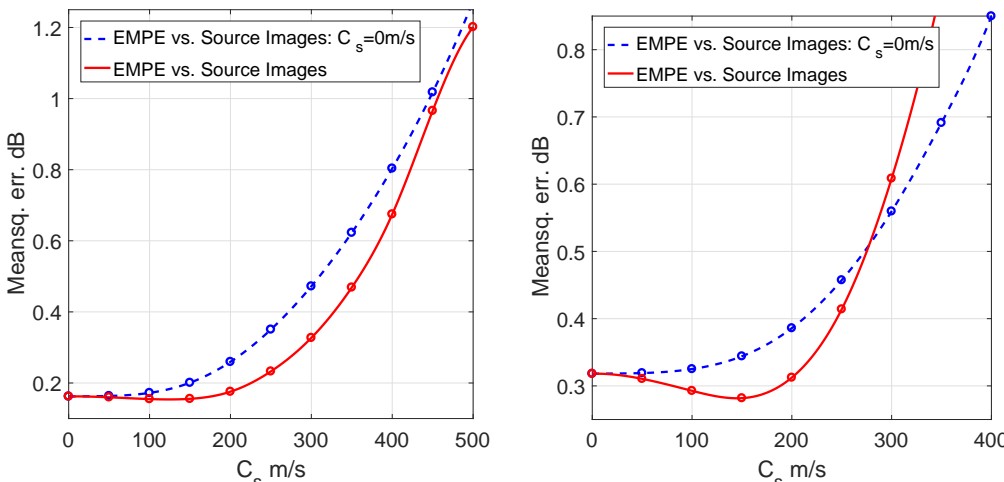

**Figure 4.** Dependence of the errors of calculations from the bottom shear speed, when $c_s$ = 0–500 m/s. Along wedge propagation for the wedge angle ≈ 2.86° (**left**) and for the wedge angle ≈ 1.14° (**right**).

In Figure 5, we investigate the influence on the accuracy of calculations by an additional small attenuation at the bottom, which could be useful for future work. Moreover, we illustrate the influence of the type of the boundary condition (Dirichlet or Neumann) at the lower boundary of the area on the accuracy of calculations.

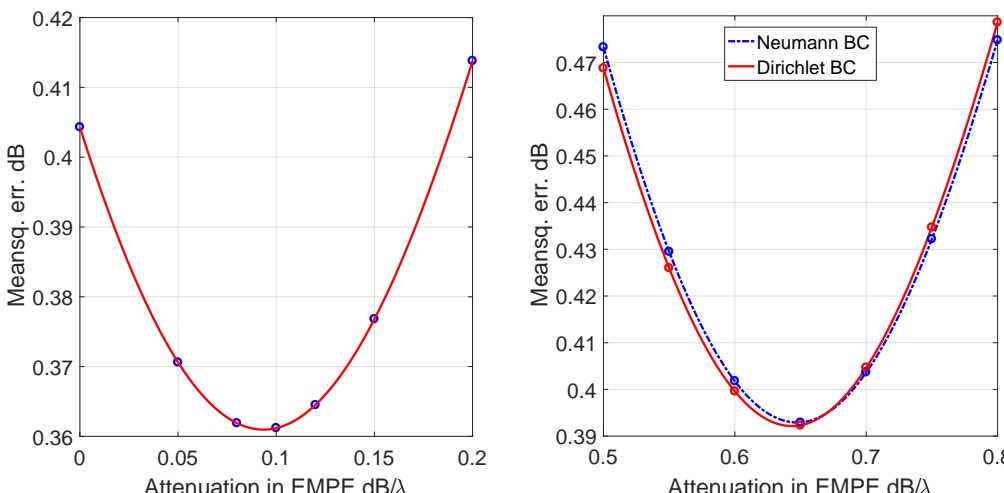

**Figure 5.** Influence on the accuracy of calculations by an additional small attenuation at the bottom for $c_s$ = 350 m/s, zero bottom attenuation (**left**), and 0.5 dB/$\lambda$ (**right**).

## 9. Discussion

In the the case of cross-wedge propagation, we discovered discrepancies at large distances between solutions obtained by the source image method and by EMPE that can be explained by the narrow-angle nature of the derived EMPE, which leads to additional errors. These errors can be significantly reduced by using wide-angle equations that take into account the elastic properties of the bottom (e.g., similar to the ones discussed in [10]). Moreover, such equations can be derived within the framework of the formalism presented in this study by considering terms arising at higher powers of a small parameter $\epsilon$. This can be a matter for future work.

The calculation time for the classical ASA wedge benchmark problem with 44 propagation modes was about 426 s on the one core of the Intel® Core™ i5-4690 K CPU, 3.50 GHz. Out of this time, 59 s were required to calculate all 44 modes along the 4 km trace with the horizontal step of 4 m (1000 steps). the vertical step was 1 m with the overall waveguide

depth $H$ = 1.5 km (1500 dots). For calculations of modes, we used the method of the inverse iteration with the first order Richardson extrapolation to increase the accuracy. To calculate the wavenumbers along the trace, the bisection algorithm was used. Of course mode calculations can be significantly accelerated with the use of parallel computing. The EMPE System (14) was solved numerically by the Crank–Nicholson implicit difference scheme in combination with the Gauss–Seidel iteration method. The step on variable $y$ was 3 m, with the width of the waveguide being 3 km, so the calculation grid was $1000 \times 1000$ nodes. To prevent reflections from the boundaries, we used transparent boundary conditions at $y = \pm 1500$ m. As transparent boundary conditions, we used the perfectly matched layers from [28]. Reflections from the bottom were suppressed by the artificially absorbing layer beginning from the depth of 1 km. The time of calculation for the EMPE system was about 367 s. Unfortunately, at this stage, calculations are hard to parallelize.

The above examples show that the elastic mode parabolic Equation (14), despite the relatively small influence of the shear modulus, has a solution different from the purely acoustic case. The derived equations are applicable in our test cases up to a shear velocity $\approx$300 m/s.

In our opinion, the MPEs used for the case of a weak elastic bottom developed in this study are the most computationally efficient tool for solving 3D sound propagation problems in shallow-water acoustics. This opinion may be justified by the fact that MPEs can be considered a result of model order reduction procedure as compared with 3D parabolic equations known from the literature (see, e.g., [29–31] and references therein).

Although one should expect that sound propagation models based on ray theory or Gaussian beams are even more efficient [3,4], they can be somewhat less accurate for low frequencies and relatively small water depth.

## 10. Conclusions

- In this work, with the use of the approach in articles [9,13,24], a mode parabolic equation method for resonantly interacting modes accounting for weak elasticity at the bottom was developed.
- The proposed method is numerically validated. The test calculations carried out for the ASA wedge benchmark prove to be in excellent agreement with the source image method [14] for shear wave speeds up to $c_s \approx 300$ m/s at the bottom and a rather good agreement of up to $c_s \approx 400$ m/s.
- We have developed the software package [32] for the modeling of sound propagation in 3D waveguides based on the derived equations.
- This software package has been successfully used to plan and analyze the results of the acoustic experiments on the propagation of sound in a shallow sea [33].

**Funding:** The work was supported by the Russian Science Foundation (project no. 22-11-00171), while the numerical experiments were conducted using a high-performance computing cluster at the Il'ichev Pacific Oceanological Institute operating in the framework of the state assignment program No. 121021700341-2).

**Institutional Review Board Statement:** Not applicable.

**Data Availability Statement:** Data sharing not applicable.

**Conflicts of Interest:** The author declares no conflict of interest.

## Appendix A. Derivation of the Main Ansatz

Here, we derive ansatz for our problem from the general form of displacement component expansion.

$$E_{\text{ef}} = \lambda + 2\mu, \quad E_{\text{ef}} = E_0(X,z) + \epsilon E_1(X,Y,z), \quad \mu = \epsilon \mu_1(X,Y,z),$$

$$\frac{1}{\rho} = \gamma_0(X,z) + \epsilon \gamma_1(X,Y,z),$$

$$\mathbf{U} = \sum_{j=0}^{\infty} \epsilon^{j/2} \mathbf{U}_{j/2}(X, Y, z, \eta, \zeta),$$

where $\mathbf{U} = (U, V, W)$ is the vector of displacements, $\eta = \xi(X, Y, z)/\sqrt{\epsilon}$, and $\zeta = \theta(X, Y, z)/\epsilon$. Now, we expand the equations of elasticity and consider terms at the different powers of $\epsilon$.

At $O(\epsilon^{-2})$, $O(\epsilon^{-3/2})$, we have:

$$W : O(\epsilon^{-2}) : \gamma_0 E_0 W_{0,\zeta\zeta} \theta_z^2 = 0,$$
$$V : O(\epsilon^{-3/2}) : \gamma_0 E_0 W_{0,\zeta\zeta} \theta_Y \theta_z = 0.$$

To satisfy the equations at $\epsilon^{-2}$ and at $\epsilon^{-3/2}$, we can choose $\theta_z = 0$; thus, now $\theta = \theta(X, Y)$.

At $O(\epsilon^{-1})$, $O(\epsilon^{-1/2})$, we have:

$$V : O(\epsilon^{-1}) : \gamma_0 E_0 (V_{0,\zeta\zeta} \theta_Y + W_{0,\eta\zeta} \xi_z) \theta_Y = 0$$
$$W : O(\epsilon^{-1}) : \gamma_0 E_0 (V_{0,\eta\zeta} \theta_Y + W_{0,\eta\eta} \xi_z) \xi_z = 0,$$
$$U : O(\epsilon^{-1/2}) : \gamma_0 E_0 (V_{0,\zeta\zeta} \theta_Y + W_{0,\eta\zeta} \xi_z) \theta_X = 0.$$

From the equation at $\epsilon^{-1}$ for $V$, we can conclude that $\theta_Y = 0$. Consequently $\theta = \theta(X)$, and note that shear waves essentially depend on $\xi$, and $\xi_z \neq 0$ in all cases. At $\epsilon^{-1/2}$ in the equation for $U$ and at $\epsilon^{-1}$ in equation for $W$, we obtain $W_{0,\eta} = 0$, so $W_0 = w_0(X, Y, z, \zeta)$.

$$W : O(\epsilon^{-1/2}) : \gamma_0 E_0 (U_{0,\eta\zeta} \theta_X + W_{1/2,\eta\eta} \xi_z + V_{0,\eta\eta} \xi_Y) \xi_z = 0$$
$$V : O(\epsilon^0) : \gamma_0 E_0 (U_{0,\eta\zeta} \theta_X + W_{1/2,\eta\eta} \xi_z + V_{0,\eta\eta} \xi_Y) \xi_Y$$
$$+ \gamma_0 \mu_1 V_{0,\eta\eta} \xi_z^2 + \omega^2 V_0 = 0.$$

From the equation for $W$ at $\epsilon^{-1/2}$, we can conclude that $U_{0,\eta\zeta} \theta_X + W_{1/2,\eta\eta} \xi_z + V_{0,\eta\eta} \xi_Y = 0$. Using this result, from the equation for $V$ at $\epsilon^0$, we have $\gamma_0 \mu_1 V_{0,\eta\eta} \xi_z^2 + \omega^2 V_0 = 0$, i.e., the Helmholtz equation for the elastic shear waves. However, in our problem, we generate by the point source only compressional waves in the liquid, without any independent shear waves in the bottom. So, we can postulate that $V_0 = 0$. Thus, we drop out the solutions with dominating shear waves. Now, we should require

$$U_{0,\eta\zeta} \theta_X + W_{1/2,\eta\eta} \xi_z = 0.$$

To resolve the equations at different powers of $\epsilon$ with respect to $\zeta$, we can seek variables $U_j$ in the form

$$U_j(X, Y, z, \eta, \zeta) = \tilde{U}_j(X, Y, z, \eta) \exp(i\zeta), \quad j = 0, 1/2, 1, 3/2, \dots.$$

This is also true for $V_j$ and $W_j$, and further tildes are omitted. Integrating the above equation by $\eta$, we obtain $U_0 = -W_{1/2,\eta} \xi_z / (i\theta_X) + u_0(X, Y, z)$, where $u_0(X, Y, z)$ is the integration constant. Denote $u_{S0}(X, Y, z, \eta) = -W_{1/2,\eta} \xi_z / (i\theta_X)$ and integrate this equality by $\eta$. As a result, we have

$$W_{1/2} = w_{S1/2}(X, Y, z, \eta) + w_{1/2}(X, Y, z),$$

where $w_{1/2}$ is also the integration constant and $w_{S1/2,\eta} = -i\theta_X u_{S0} / \xi_z$.

At $O(\epsilon^0), O(\epsilon^{1/2})$, we have:

$$U : O(\epsilon^0) : i\theta_X \gamma_0 E_0 (w_{0,z} + i\theta_X U_0) + i\theta_X \gamma_0 E_0 W_{1/2,\eta} \xi_z$$
$$+ \gamma_0 \mu_1 U_{0,\eta\eta} \xi_z^2 + \omega^2 U_0 = 0$$
$$W : O(\epsilon^0) : \gamma_0 [E_0 (w_{0,z} + i\theta_X u_0)]_z$$
$$+ \gamma_0 E_0 (i\theta_X U_{1/2,\eta} + W_{1,\eta\eta} \xi_z + V_{1/2,\eta\eta} \xi_Y) \xi_z + \omega^2 w_0 = 0$$
$$V : O(\epsilon^{1/2}) : \gamma_0 E_0 (w_{0z} + i\theta_X u_0)_Y + \gamma_0 \mu_1 V_{1/2,\eta\eta} + \omega^2 V_{1/2}$$
$$+ \gamma_0 E_0 (i\theta_X U_{1/2,\eta} + W_{1,\eta\eta} \xi_z + V_{1/2,\eta\eta} \xi_Y) \xi_Y = 0.$$

The consideration of the equations for $U$ at $\epsilon^0$ gives two equations:

$$i\theta_X \gamma_0 E_0 (w_{0z} + i\theta_X u_0) + \omega^2 u_0 = 0, \quad \gamma_0 \mu_1 u_{S0\eta\eta} \xi_z^2 + \omega^2 u_{S0} = 0. \tag{A1}$$

The last equation is the Helmholtz equation for the SV elastic shear waves. However, in our problem, we can postulate $u_{S0} = 0$ and also $w_{S1/2} = 0$. Thus, we drop out solutions with dominating shear waves.

The consideration of the equations for $W$ at $\epsilon^0$ also gives two equations:

$$\gamma_0 [E_0 (w_{0z} + i\theta_X u_0)]_z + \omega^2 w_0 = 0, \quad i\theta_X U_{1/2,\eta} + W_{1,\eta\eta} \xi_z + V_{0,\eta\eta} \xi_Y = 0. \tag{A2}$$

The last equation here eliminates the last term in the equation for $V$ at $\epsilon^{1/2}$. Thus, from the equation for $V$ at $\epsilon^{1/2}$, we obtain two separate equations:

$$\gamma_0 E_0 (w_{0z} + i\theta_X u_0)_Y + \omega^2 v_{1/2} = 0, \quad \gamma_0 \mu_1 v_{S1/2,\eta\eta} \xi_z^2 + \omega^2 v_{S1/2} = 0.$$

The equation for $V$ at $\epsilon^{1/2}$ thus (regarding the above results) leads to $V_{1/2} = v_{1/2}(X, Y, z) + v_{S1/2}(X, Y, z, \eta)$, where $\omega^2 v_{1/2} = -\gamma_0 E_0 (w_{0z} + i\theta_X u_0)_Y$, and $\gamma_0 \mu_1 v_{S1/2,\eta\eta} + \omega^2 v_{S1/2} = 0$. Again, the last equation is the Helmholtz equation for the SH elastic shear waves. However, in our problem, any independent shear waves in the main order are absent. So, we can postulate that $v_{S1/2} = 0$. Moreover, from the second equation in (A2), we can conclude that $U_{1/2} = u_{1/2}(X, Y, z) + u_{S1/2}(X, Y, z, \eta)$ and $W_1 = w_{S1}(X, Y, z, \eta) + w_1(X, Y, z)$, where $w_{S1,\eta} = -i\theta_X u_{S1/2} / \xi_z$.

Introduce value $P_0 = -E_0 (w_{0,z} + i\theta_X u_0)$. From the first equations in (A1) and (A2), we obtain the equation for $P_0$:

$$(\gamma_0 P_{0,z})_z + \gamma_0 \left( \frac{\omega^2}{\gamma_0 E_0} - \theta_X^2 \right) P_0 = 0.$$

With the appropriate boundary and interface conditions for $P_0$, this equation consists of a typical acoustic Sturm–Liouville problem, with the spectral parameter $\theta_X^2$ depending on the slow variable $X$.

From the equations for $U$ at $\epsilon^{1/2}$, we have two equations:

$$i\theta_X \gamma_0 E_0 (w_{1/2,z} + i\theta_X u_{1/2}) + \omega^2 u_{1/2} = 0, \quad \gamma_0 \mu_1 u_{S1/2,\eta\eta} \xi_z^2 + \omega^2 u_{S1/2} = 0.$$

Additionally, from the equation for $W$ at $\epsilon^{1/2}$, we have:

$$\gamma_0 [E_0 (w_{1/2,z} + i\theta_X u_{1/2})]_z + \omega^2 w_{1/2} = 0.$$

Combining the above equations and introducing value $P_{1/2} = -E_0 (w_{1/2,z} + i\theta_X u_{1/2})$, we obtain the equation for $P_{1/2}$:

$$(\gamma_0 P_{1/2,z})_z + \gamma_0 \left( \frac{\omega^2}{\gamma_0 E_0} - \theta_X^2 \right) P_{1/2} = 0.$$

This equation exactly coincides with the one for $P_0$. Without loss of generality, we can put $P_{1/2} = 0$, and also $w_{1/2} = 0$ and $u_{1/2} = 0$. Because $w_{1/2}$, $u_{1/2}$, and, related with them, $v_1$ are absent in the equations for $U$, $V$ at $\epsilon^1$, we can introduce the following representation of displacements, which gives us the final parabolic equation.

$$
\begin{aligned}
u = {}& [u_0(X,Y,z) + \epsilon u_1(X,Y,z)]e^{i\theta(X)/\epsilon} \\
& + [\epsilon^{1/2}u_{S1/2}(X,Y,z,\eta) + \epsilon u_{S1}(X,Y,z,\eta)]e^{i\theta(X)/\epsilon} + \dots, \\
v = {}& [\epsilon^{1/2}v_{1/2}(X,Y,z) + \epsilon^{3/2}v_{3/2}(X,Y,z)]e^{i\theta(X)/\epsilon} \\
& + [\epsilon v_{S1}(X,Y,z,\eta) + \epsilon^{3/2}v_{S3/2}(X,Y,z,\eta)]e^{i\theta(X)/\epsilon} + \dots, \\
w = {}& [w_0(X,Y,z) + \epsilon w_1(X,Y,z)]e^{i\theta(X)/\epsilon} \\
& + [\epsilon w_{S1}(X,Y,z,\eta) + \epsilon^{3/2}w_{S3/2}(X,Y,z,\eta)e^{i\theta(X)/\epsilon} + \dots.
\end{aligned}
\tag{A3}
$$

Thus, the obtained ansatz contains two groups of members: describing compressional waves in the water and at the bottom and describing shear waves at the bottom.

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
