# Peer review of "Coupled-Mode Parabolic Equations for the Modeling of Sound Propagation in a Shallow-Water Waveguide with Weak Elastic Bottom"

_jmse, doi:10.3390/jmse10101355_

Round 1
Reviewer 1 Report
Excellent manuscript that can be published with only very minor updates.
Just a few comments:
1) In the expansion (2) and the WKB ansatz i am missing some arguments, why this is reasonable or justified. The author simply "postulates". Are there any physical arguments / assumptions for this procedure? This would also shed light on the question in which parameter regimes this approach is feasible.
2) developped Software: it is left unclear whether this software is available,
e.g. on gitlab or github.
3) References : please use a unique style, i.e. all journal names have to be abbreviated.
Author Response
1) In the expansion (2) and the WKB ansatz i am missing some arguments,
why this is reasonable or justified. The author simply "postulates".
Are there any physical arguments / assumptions for this procedure?
This would also shed light on the question in which parameter regimes
this approach is feasible.
> I have inserted 3 paragraphs at the top of section "WKB solutions for shear waves" with physical arguments. The word "postulate" was removed.
2) developped Software: it is left unclear whether this software is available,
e.g. on gitlab or github.
> This software is locally developed on C++ and can be downloaded at the address https://disk.yandex.ru/d/Ao2lyJQBuNB5pQ This address was added to literature list. In some future it can be uploaded on github when its help description is ready.
3) References : please use a unique style, i.e. all journal names have to be abbreviated.
> Corrected.
Author Response
1. The introduction section is too simplistic. For this kind of comprehensive journal (non-ocean acoustics is only a small direction), it is very necessary for background introduction and popular science. Therefore, I do not recommend introducing coupled-mode parabolic equations right away.
> The introductory section has been significantly expanded.
2. The author should first introduce the common models in computational ocean acoustics (rays, wavenumber integral, normal modes, and parabolic models). Then the combination (MPE) of the two is derived from the normal modes/coupled modes and parabolic models.
> Done.
3. The literature survey in the Introduction is insufficient. For NM and PE, many new methods have been proposed in recent years [1–6], and the authors should include them in this manuscript to make the literature list trendy enough. (red means important! )
> All the recommended articles have been included in the literature list.
4. The meaning of ? in Eq. 1 is not explained
> Corrected.
5. Why does the P1 after formula 5 wrap?
> Corrected.
6. The author does not state how the weak elasticity in the title should be quantified? Although the cs in the simulation example can be used as a simple reference.
> The discussion of the weak elasticity in the 5th paragraph of the "Introduction" is added.
7. The simulation is too simple and only considers the wedge-shaped waveguide ASA example. Can his model handle other terrains?
> Of course, this model can handle any other terrains where horizontal changes
of bathymetry and hydrology are slow in comparison with the wavelendth.
The 1st paragraph in the section "Numerical calculations" was added.
8. I am more concerned about the speed of 3D sound field calculation, can the author show me their speed? Of course, as an ordinary paper, the authors may not introduce a discussion of computational efficiency within the manuscript.
> 2nd paragraph in the added "Discussion" section is devoteed to computational
efficiency of the EMPE method.
9. The structure of Section 9 of the manuscript is very unreasonable, and the last
two paragraphs are clearly discussing the advantages and disadvantages of various methods, which should be placed in the Discussion.
> Section "Discussion" is added to manuscript.
10. The manuscript shows the author’s solid theoretical foundation. But as far as essay writing is concerned, it is not well organized. The author should take care of the logic and structure of the manuscript and polish the text. I still agree that the manuscript is published after an overhaul. Looking forward to seeing this manuscript get better.
> The manuscript was reorganized by adding "Discussion" section where some paragraphs from other sections was placed.
Round 2
Reviewer 2 Report
I am pleased to see that after a round of revisions the quality of the manuscript has improved significantly and I recommend it for publication. However, I think ten sections are too many. It is recommended to combine them if they can. Seven sections are more suitable.